# Gastrointestinal Tract Stabilized Protein Delivery Using Disulfide Thermostable Exoshell System

**DOI:** 10.3390/ijms23179856

**Published:** 2022-08-30

**Authors:** Samira Sadeghi, Girish Vallerinteavide Mavelli, Siddhesh Sujit Vaidya, Chester Lee Drum

**Affiliations:** 1Yong Loo Lin School of Medicine, 14 Medical Drive, National University of Singapore, Singapore 117599, Singapore; 2Genome Institute of Singapore, Agency for Science, Technology and Research (A*Star), Singapore 138672, Singapore

**Keywords:** gastrointestinal, thermostable exoshells, DS-tES, tES, HRP, rluc

## Abstract

Thermostable exoshells (tES) are engineered proteinaceous nanoparticles used for the rapid encapsulation of therapeutic proteins/enzymes, whereby the nanoplatform protects the payload from proteases and other denaturants. Given the significance of oral delivery as the preferred model for drug administration, we structurally improved the stability of tES through multiple inter-subunit disulfide linkages that were initially absent in the parent molecule. The disulfide-linked tES, as compared to tES, significantly stabilized the activity of encapsulated horseradish peroxidase (HRP) at acidic pH and against the primary human digestive enzymes, pepsin, and trypsin. Furthermore, the disulfide-linked tES (DS-tES) exhibited significant intestinal permeability as evaluated using Caco2 cells. In vivo bioluminescence assay showed that encapsulated Renilla luciferase (rluc) was ~3 times more stable in mice compared to the free enzyme. DS-tES collected mice feces had ~100 times more active enzyme in comparison to the control (free enzyme) after 24 h of oral administration, demonstrating strong intestinal stability. Taken together, the in vitro and in vivo results demonstrate the potential of DS-tES for intraluminal and systemic oral drug delivery applications.

## 1. Introduction

Oral is the most preferred and convenient mode among all drug administration routes and is the dominant mode of drug administration for common medicines. Oral delivery offers the best patient compliance as drugs are self-administered with flexible dosages following the patients’ needs [1,2,3]. In addition, the gastrointestinal (GI) tract mucosa and epithelial cells exhibit a large surface area of 30–40 m^2^ and thus the opportunity for high mass transfer and absorption. Goblet cells, endocrine cells, Paneth cells, and specialized M cells on the GI tract may likewise perform specialized roles in drug adsorption offering unique characteristics for oral drug development [4,5]. Although a standard for small molecule administration routes, the field of nanomedicine has yet to find a material base for consistent oral drug delivery due to the often delicate superstructures required of biocompatible polymers and lipids in the context of high acidity and unfavorable physicochemical properties of the GI tract. Ultimately, nanoparticle delivery technologies remain with poor bioavailability and a need for alternative materials which are stable throughout the GI tract [1,6,7].

Proteins and peptides are widely used in clinical care with only <5% administered orally, mostly due to the instability at pH variations and degradative enzymes in the stomach and proximal small intestine [8]. Nanoparticle systems have improved the oral delivery of small molecule drugs, but relatively little progress has been made using functionally active proteins or enzymes, owing to the incompatibility of organic emulsions used for the synthesis of these systems and the tertiary structures of proteins required for activity. Thus, the combination of denaturation and low encapsulation efficiency results in an ongoing challenge [9,10,11]. Highly stabilized nanoparticles are a promising approach toward biocompatibility, biodegradability, non-immunogenicity, resistance to enzymatic degradation, and amenability to genetic modifications for surface engineering [12,13,14]. One such approach is the self-assembled and monodisperse protein cage, ferritin, engineered to have enhanced circulation half-life and low toxicity [14,15,16]. Currently, however, gastric pH causes these protein cages to disassemble, exposing the therapeutic protein cargo and leading to activity loss and degradation by digestive enzymes [1,17,18,19]. Further, these cages have to traverse through the thick mucus layer on the GI tract to reach the lumen, limiting their application as an oral delivery vehicle [7,20].

We recently developed a novel protein cage ‘tES’ by engineering the thermostable *Archaeoglobus fulgidus* ferritin. They are 12 nm in diameter with an 8 nm aqueous cavity, a volume that can theoretically accommodate protein(s) of molecular volume ~270 nm^3^. The shells contain four 4.5 nm surface pores, which allow small molecule substrates to permeate while blocking the interaction of macromolecules such as antibodies and proteases [21]. The advantages of tES are: (1) simple and rapid encapsulation of protein cargo using charge complementation and mild pH titrations, (2) prevention of aggregation, aiding in proper tertiary structures and maintenance of protein activity, and (3) highly stabilized and protective environment for internalized proteins against denaturation and proteolysis, making it an ideal vector for enzyme delivery. Additionally, tES has been shown to function as an artificial chaperone to fold proteins in vitro [22], stabilize labile proteins from freezing and drying stresses during lyophilization [23], and serve as a multilayered catalytic center for bioorthogonal catalysis to treat solid tumors [24].

Here, we describe the structural refinement of tES by incorporating multiple inter-subunit disulfide linkages to create a disulfide-linked tES with practicable advantages in oral delivery applications. We observed that by engineering natural covalent crosslinking using molecularly precise disulfide linkages, ie DS-tES, we were able to protect encapsulated enzymes against the adverse GI environment when compared to tES and/or enzyme alone. Furthermore, an in vitro Caco2 assay exhibited permeabilization through intestinal epithelial cells, confirming enhanced absorption. Finally, rluc encapsulated within DS-tES displayed clear protein stabilization throughout all sections of the GI tract via the clear presence of bioluminescence in feces following oral administration in mice. To the best of our knowledge, this is the first report of an engineered protein shell with applications in the oral delivery of protein cargos. 

## 2. Results

### 2.1. Design and Screening for Disulfide tES 

Previously, we reported the application of tES in delivering an iron-mediated catalytic center for intratumoral bioorthogonal catalysis and regression of solid tumors [24]. To extend the potential applications for tES to additional delivery routes, we improved the stability of tES by introducing engineered disulfide bonds at subunit interfaces. The tES subunit amino acid sequence was first screened to find residues with ~2 Å interatomic distance. We found that arginine 151(R151), alanine 152(A152), leucine 53(L53), alanine 74(A74), glycine 76(G76), arginine 66(R66), glycine 67(G67), glutamine 50(Q50), glycine 37 (G37), and alanine 117(A117) are located in the tES dimerization intersection and thus designed mutagenesis primers to systematically substitute them to cysteine. As observed in non-reducing SDS-PAGE, R151C did not form disulfide bonds between the subunits whereas A152C formed dimers (Appendix A). Subsequent double mutants (A152C-L53C and A152C-A74C) exhibited a higher propensity for disulfide bond formation between the subunits in comparison to A152C-G76C (Appendix A). Hence, the two variants were selected for further mutagenesis to form the double/triple mutants (A152C-R66C; A152C-G67C; A152C-L53C-R66C; A152C-A74C-R66C; A152C-L53C-G67C; and A152C-A74C-G67C) (Appendix A). Of these, A152C-G67C and A152C-L53C-G67C improved the stability on the non-reducing SDS-PAGE, though the full linked shell was not observed (Appendix A). Diverse disulfide induction methodologies such as the use of dimethyl sulphoxide [25], glutathione system (GSH-GSSG) [26,27], methanol [28], hydrogen peroxide [29], iodine [30], BMOE cross-linker [31], and air oxidation in diverse pH were adopted. However, none of these resulted in the formation of fully linked and stabilized tES (Appendix A). Further screening revealed the significance of G37C and A117C in stabilizing tES in a non-reducing environment (Appendix A). Therefore, the A152C-G67C and A152C-L53C-G67C tES were mutated to their corresponding constructs, A152C-G67C-A117C-G37C and A152C-L53C-G67C-A117C-G37C, and evaluated for their stability at pH 4.0. Interestingly, A152C-L53C-G67C-A117C-G37C (DS-tES) exhibited higher stability at the acidic pH and was selected for further studies (Figure 1a–c). All the clones were sequence confirmed, mutants expressed, and proteins purified and evaluated in a non-reducing SDS-PAGE.

### 2.2. Purification and Characterization of DS-tES

As indicated in Figure 2a,b, mutagenesis at the five target sites (A152C-L53C-G67C-A117C-G37C) contributed to the DS-tES and reduction with TCEP suggested that DS-tES is more stable than tES in a non-reducing environment even in the presence of denaturants (Figure 2a,b and Appendix A). DS-tES has hydrodynamic diameter of ~13 nm and elution of 10 mL in size exclusion chromatography (Figure 2c,d). Moreover Compared to tES, which dissociates into monomers at pH 4.0, DS-tES remained fully assembled and folded in extremely acidic pH (Figure 1c). Thiolation is an efficient method for customization of the mucoadhesive and mechanical properties of drug delivery systems [32,33], we thus quantified the thiol group content in the DS-tES using Ellman’s reagent and determined it to be 1.2 micromoles per mg of protein (Appendix A). Further, the endotoxin levels in the purified protein samples were estimated to be 0.018 ng/mL per mg of protein and were within the limits reported previously [34,35]. Overall, the cage was found to be covalently ‘locked’ and adopt an extremophile stability profile. This new quality is the result of 60 rationally engineered disulfide bonds that stabilize intersubunit interactions without affecting the overall morphology of tES [21,22].

### 2.3. DS-tES Stabilized HRP Activity in Simulated GI Conditions

Acidic gastric fluid, together with enzymatic barriers such as pepsin and trypsin, is biologically designed to degrade proteins and thus prohibits the administration of most protein-based therapeutics. We thus evaluated the stability of HRP as a model peptide encapsulated within the DS-tES (DS-tES-HRP) in a simulated GI environment, and the activity was compared to free HRP or tES-HRP (non-disulfide stabilized). In the presence of pepsin and pH 4.0, DS-tES-HRP exhibited ~5 times and ~14 times more enzymatic activity compared to tES-HRP or free HRP, respectively (Figure 3a). However, when incubated with trypsin at pH 7.0, both DS-tES and tES protected the encapsulated HRP against proteolysis equally (Figure 3b). The protective property of tES towards the encapsulated protein cargo from trypsinolysis was previously reported [21]. On a similar note, tES maintained ~30% of HRP activity when incubated with serum proteases for a duration of 8-days, but with no quantifiable activity observed with free HRP [24]. Thus, DS-tES protects internalized protein substrate from the two major GI proteases in both normal and acidic pH environments.

### 2.4. DS-tES Permeabilizes through Intestinal Cell Layers

To assess transcytosis of DS-tES through intestinal epithelium, a permeability assay was carried out in human colon carcinoma cell line Caco2 as it represents the best available in vitro model of small intestine enterocytes [36,37]. As evident from Figure 3c, the HRP signal was significantly higher upon encapsulation in DS-tES at all points evaluated. DS-tES-HRP exhibited ~2.5 times more peroxidase activity in comparison to free HRP after 3 h. Moreover, a mild reduction in peroxidase activity from DS-tES to tES was observed wherein the difference could be attributed to the free thiols present on the DS-tES. For further validation, we studied the permeability of rluc encapsulated within DS-tES (DS-tES-rluc). The luciferase activity was found to be ~3 times higher compared to free rluc after 3 h. Further, a difference of ~1.5 times in luciferase activity was observed between DS-tES-rluc and tES-rluc (Appendix A). Similar such thiol-mediated permeabilization through the Caco2 monolayers was reported previously [38,39]. Taken together, these observations suggest a promising role for DS-tES in intestinal epithelial transfer. 

### 2.5. Oral Delivery of DS-tES and Its In Vivo Evaluation

As prior studies with tES-rluc exhibited diminished luciferase activity similar to the free enzyme (Appendix A), we further evaluated the in vivo capabilities of DS-tES as an oral delivery vehicle. A single dose of both rluc and DS-tES-rluc was orally administered to Balb/c mice. The ViviRen substrate was then intraperitoneally injected into mice after 3 and 24 h (Figure 4a). As can be seen in Figure 4b, both rluc and DS-tES-rluc moved through the GI tract by 3 h post-administration. Over the next 24 h, the encapsulated luciferase activity was maintained in the GI tract but negligible activity was observed with free rluc. Thereafter, the total signal decreased over time as rluc, whether free or encapsulated, was eliminated from the GI tract. Additionally, we measured the rluc activity from mice feces. A strongly significant effect of DS-tES (*p* < 0.001) was observed which protected rluc activity for the length of the GI tract up to 24 h. The luciferase activity was then reduced ~6 times from 24 h to 72 h. On the contrary, the free luciferase activity was reduced to more than 50 times after 24 h, suggesting that DS-tES could protect and maintain the luciferase activity upon oral administration (Figure 4c).

## 3. Discussion

The success of nanoparticle technology in oral administration is highly dependent on its survival in the GI environment. Protein-based nanoparticles such as gliadin [40,41], lectin [42,43], casein [44,45], and zein [44,46] have shown early progress; however, retain serious limitations. These proteins require desolvation or coacervation techniques to precipitate nanoparticles which often retain a high degree of heterogeneity [47,48]. Due to their low stability, they often require additives to stabilize their structure [40,49]. In addition, they are majorly used in loading small molecule drugs and not therapeutic proteins or enzymes [13,14]. On the contrary, both DS-tES and tES derived from the native nanoparticle exhibit homogeneous size distribution in addition to stability against physicochemical stresses and proteolytic degradation [23]. Further, their small size enables them to exit the vasculature, permeate the tissue, and or enter the lymphatic system through passive diffusion [12]. A comparative analysis of DS-tES and tES is summarized in Appendix A. Despite the availability of multiple ferritin-based drug delivery systems, so far none has been successful in oral delivery applications [16,50]. Moreover, DS-tES encapsulates and protects diverse protein cargos within their aqueous cavity, a feature absent in other ferritin-based systems where the protein therapeutics are usually carried on their surface where they get exposed to denaturing and proteolytic environment [21,22,51,52].

The transit of protein nanoparticles through the GI tract involves exposure to a strong pH gradient (pH 2.0–8.0), digestive enzymes (pepsin, trypsin, etc.), and mucosal and cellular barriers, that collectively affect the structure-function of the loaded therapeutic proteins or peptides. Our study demonstrates that DS-tES protect encapsulated substrates in this environment (Figure 3a–c). It is noteworthy that the gastric pH is maintained at 4.0–5.0 while in the fed condition, indicating the role of food intake in gastric pH [43,53]. Receptor-mediated endocytosis [54] is considered advantageous for the cellular uptake and transport across the GI epithelium and transferrin receptor 1 (TfR1) is an attractive target for drug delivery currently under evaluation [55,56,57,58]. TfR1 is a transmembrane glycoprotein with a critical role in cellular iron uptake through endocytosis of iron-binding proteins, transferrin, and ferritin [57,59]. Interestingly, TfR1 is expressed at high levels in two interesting cell types: cancer cells and the GI epithelium [60,61]. The TfR1-mediated pathway is thus an interesting target for transcytosis of drug-loaded nanoparticles in the GI tract [61,62,63,64]. Similar to the previously characterized ferritin-based systems, tES is endocytosed through receptor-mediated mechanisms [24,65,66]. Likewise, DS-tES is differentiated from tES due to the high prevalence of cysteines and disulfide bonds which stabilize the quaternary structure. Thiols may play an additional role to DS-tES in uptake, specifically thiol-mediated cellular uptake through transferrin receptors [67,68] in addition to improved mucoadhesiveness [32,68]. We also note that, when administered orally, DS-tES encapsulating rluc exhibited a long residence time within the GI tract, before getting eliminated from the system. However, DS-tES, like tES, may be limited in encapsulating therapeutics up to a molecular volume of ~270nm^3^ [22]. Additionally, the presence of free surface thiols may lead to non-specific aggregation affecting the therapeutic efficacy.

## 4. Materials and Methods

### 4.1. Preparation of DS-tES

The amino acids at the tES dimerization intersection were identified for nucleotide substitution based on interatomic distance using the PyMOL software. The PRSF plasmid carrying tES gene19 was purified using the OMEGA E.Z.N.A Plasmid DNA Mini Kit I (D6922-02). The primers for nucleotide replacement were designed using QuikChange Primer Design and the PCR reaction was performed using the QuikChange lightening Site-Directed Mutagenesis Kit (210519). The PCR product was transformed into competent cells provided by the kit. The colonies were selected using kanamycin (25 µg⋅mL^−1^; ThermoFisher, Singapore) resistance LB agar (Axil Scientific, USA) plates and subsequently sequenced for positive confirmation. The positive constructs were transformed into BL21(DE3) *E. coli*-competent cells. An overnight starter culture (25 mL) prepared from a single positive colony was used to inoculate LB broth (1000 mL) maintained at 37 °C and supplemented with kanamycin. The bacteria culture was grown until the absorbance (OD600) of ~0.7 was obtained. Protein expression was induced using isopropyl β-D-1-thiogalactopyranoside (IPTG, 0.4 mM; Axil Scientific), and the culture was allowed to grow for another 5 h at 37 °C. The bacteria were pelleted by centrifugation at 14,000× *g* for 15 min. The pellet was resuspended in the lysis buffer (50 mM Tris-HCl, pH 8.0, 200 mM NaCl, 5 mM BME, and 0.1% Triton-X 100) and incubated on ice for 15 min, followed by sonication and centrifugation. As previously described [21], the DS-tES was purified under reducing conditions. The purity of the proteins was evaluated using sodium dodecyl sulfate-polyacrylamide gel electrophoresis (SDS-PAGE) and stored under reducing conditions for long-term usage.

### 4.2. Characterization of DS-tES

The particle sizes of tES and DS-tES were measured by dynamic light scattering (DLS) studies (Nanobrook Omni, Brookhaven). The purified tES and DS-tES were aliquoted in sized cuvettes and the measurements were achieved at a scattering angle of 90°. The formation of disulfide bonds was assessed using non-reducing SDS-PAGE. In addition, the concentration of free thiols present in the DS-tES was determined using Ellman’s Assay. Briefly, 5,5′-dithio-bis-2-(nitrobenzoic acid) or DTNB prepared at 4 mg/mL concentration in 0.1 M sodium phosphate buffer was mixed with protein samples and the generation of the yellow adduct was measured at 412 nm. A standard graph was determined using known concentrations of L-cysteine and the free thiols were quantified. Additionally, the presence of endotoxin in the purified protein samples was determined using an endotoxin kit (Thermo Fisher Scientific, Waltham, MA, USA) as per the user instructions.

### 4.3. Preparation of DS-tES-HRP/rluc

The DS-tES in presence of the reducing agent was acidified to pH 5.8 to obtain the subunits. The subunits were mixed with a 10-fold molar excess of HRP (Thermo Fisher Scientific, Waltham, MA, USA), and incubated at 4 °C for 30 min. The pH of the mixture was increased to 8.0 and the assembled DS-tES encapsulating HRP was separated from free HRP using a non-reducing buffer on SEC. The HRP encapsulation was analyzed using SDS-PAGE. Contrarily, DS-tES-rluc was prepared as described earlier with the above-mentioned nucleotide substitutions [21,24].

### 4.4. In Vitro Experiments

#### 4.4.1. Pepsin Assay

DS-tES-HRP/tES-HRP/HRP were incubated for 2h in the presence of 1 mg/mL pepsin. The molar ratio of HRP to pepsin used in this study was 1:100 (for DS-tES-HRP and tES-HRP the equivalent HRP activity as free HRP was used) and 3 M HCl was used to make different pH values ranging from 2 to 8 and pH was reverted to 8 by NaOH to deactivate the pepsin enzyme. The HRP activity was assayed after 2 h using 3,3′,5,5′-tetramethylbenzidine (TMB) substrate in a 96-well crystal-clear polystyrene plate (Greiner Bio-One). Absorbance was measured at 450 nm using a Perkin plate reader.

#### 4.4.2. Trypsin Assay

DS-tES-HRP/tES-HRP/HRP were incubated for 2 h in different concentrations of trypsin (0.25 and 0.4%). The HRP activity was measured similarly to as described in the pepsin assay.

#### 4.4.3. Cell Permeability

The penetrance of the DS-tES into the epithelium of the intestine was carried out in the human colon carcinoma cell line, Caco-2. Briefly, 0.5 million Caco2 cells were seeded on transwell plates and incubated in a 10% CO_2_ incubator for 21 days at 37 °C. The permeability assay was performed as described before [69]. The HRP and rluc activity was determined from both apical and basal sides of each filter using the formula:P_app_ = (dQ/dt) × (1/(AC_0_))

### 4.5. In Vivo Experiments

To investigate the stability of DS-tES and tES in vivo, 0.8 mM of DS-tES-rluc, tES-rluc, or equivalent activity of free rluc were orally administered to Balb/c mice. Followed by intraperitoneal injection of the ViviRen™ In Vivo Substrate (0.4 mM) in 3 and 24 h. The distribution of the bioluminescence was evaluated using an IVIS Spectrum. In addition, mice feces were collected to assess the presence of the enzyme in the GI tract after 5, 24, 48, 72 h, and 1 week of oral administration of DS-tES-rluc.

## 5. Conclusions

DS-tES may be useful in oral delivery systems which encapsulate and protect protein cargos from functional destabilization. DS-tES utilize a simple and rapid encapsulation technique without affecting the properties of the encapsulate, which may have significant implications for the nutraceutical and pharmaceutical industries. Future evaluation for suitable therapeutic formulations, systemic distribution, and toxicity studies is thus justified.

## Figures and Tables

**Figure 1 ijms-23-09856-f001:**
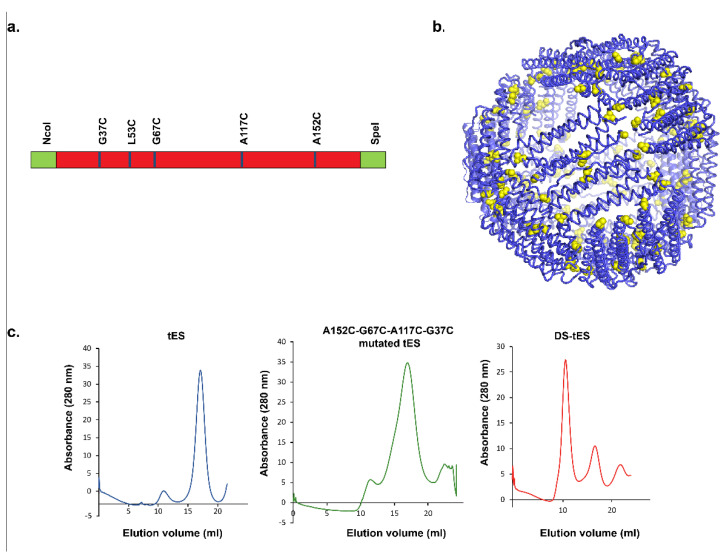
(**a**) Schematic representation of DS-tES construct with six point mutations and restriction sites. (**b**) PyMOL representation of DS-tES with all the point mutations (yellow represents the cysteine residues). (**c**) Size-exclusion chromatography of tES, A152C-G67C-A117C-G37C mutated tES, and DS-tES at pH 4.0.

**Figure 2 ijms-23-09856-f002:**
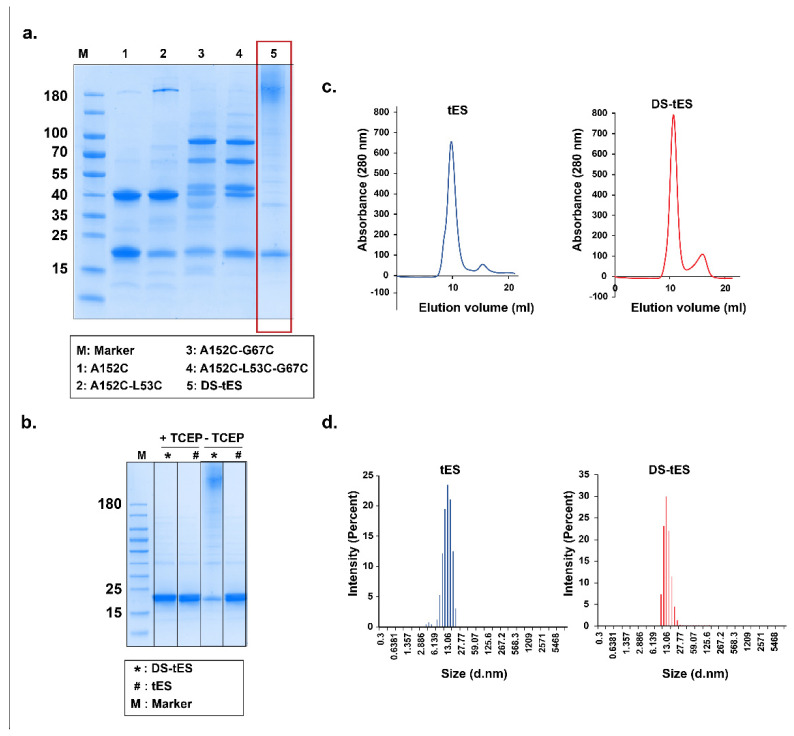
(**a**) Sequential engineering of DS-tES. Non-reducing SDS-PAGE of the selected mutants. (**b**) SDS-PAGE analysis of DS-tES and tES in the absence or presence of TCEP. (**c**) Size-exclusion chromatography of tES and DS-tES with similar elution profiles. (**d**) Hydrodynamic diameter measurements of tES and DS-tES using dynamic light scattering studies.

**Figure 3 ijms-23-09856-f003:**
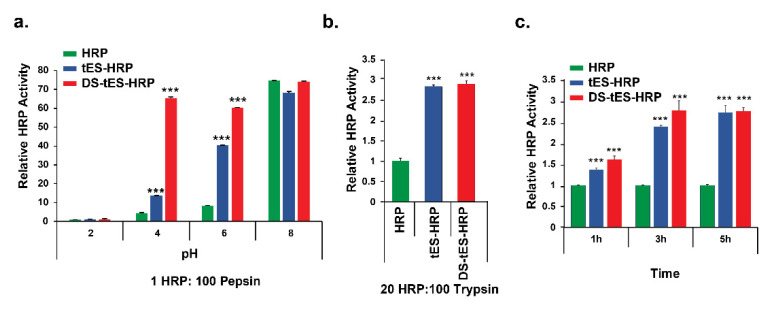
(**a**) Pepsin assay under different pH conditions. DS-tES protected the encapsulated HRP from the simulated gastric condition. (**b**) Trypsin assay, tES and DS-tES protect the HRP against trypsin enzyme. (**c**) Permeability assay in Caco2 monolayers shows that both tES and DS-tES can permeabilize through intestinal epithelium. ((**a**–**c**): data are shown as mean ± SEM, n = 3. *** *p* < 0.001.).

**Figure 4 ijms-23-09856-f004:**
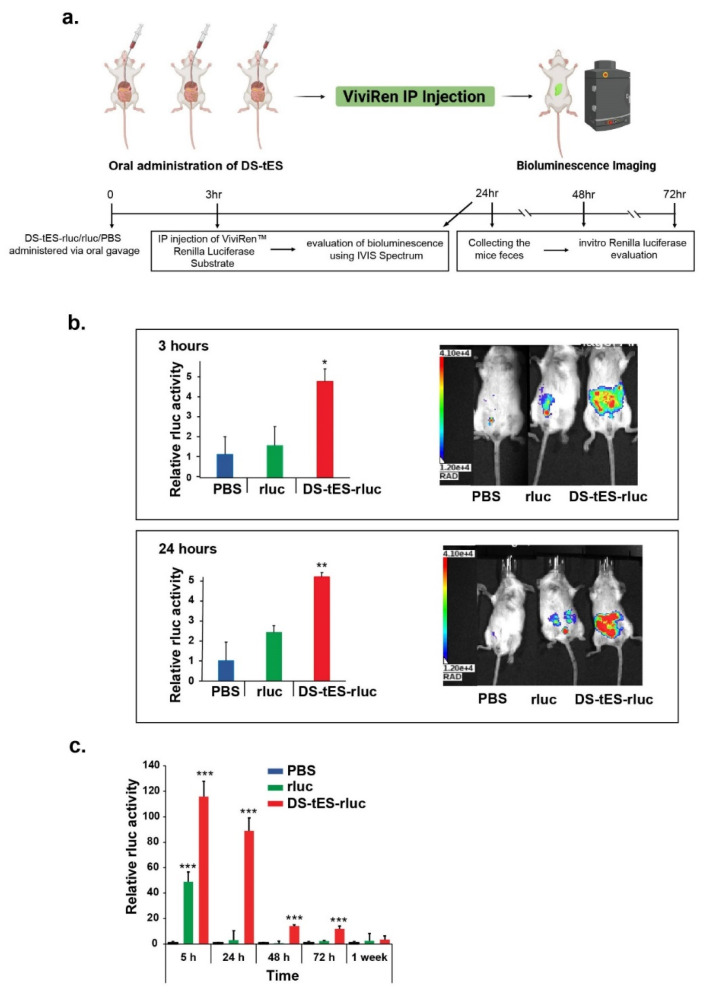
(**a**) Schematic representation of oral administration of DS-tES-rluc/rluc followed by IP injection of ViviRen substrate. (**b**) Bioluminescence emitted from mice administered with DS-tES-rluc/rluc in 3 and 24 h. (**c**) Luciferase activity measurement from mice fecal matter for up to 1 week. ((**b**,**c**): data are shown as mean ± SEM, n = 4. *** *p* < 0.001, ** *p* < 0.01, * *p* < 0.05).

## Data Availability

The data presented in this study are available on request from the corresponding author.

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
