# Peer review of "Gastrointestinal Tract Stabilized Protein Delivery Using Disulfide Thermostable Exoshell System"

_ijms, 2022, doi:10.3390/ijms23179856_

Round 1
Reviewer 1 Report
The authors reported gastrointestinal tract stabilized protein delivery using the Disulfide tES system. The submission can be accepted after revision considering the following points:-
1. The title should be revised to be informative and precise. Abbreviation,
Thermostable exoshells (tES), should be fully defined.
2. The novelty of the study should be highlighted compared to the author’s previous studies, such as Nature communications 2021, 12 (1), 1-15; Nature communications 2017, 8, 1-8.
3. A comparison with the previously published system should be discussed and summarized in a Table.
4. References should be updated.
5. The language should be revised, and typos should be corrected.
Author Response
We thank the reviewer #1 for constructive comments to make the manuscript more readable and appealing to a larger scientific population. The responses to the specific comments are given below:
1- The title should be revised to be informative and precise. Abbreviation, Thermostable exoshells (tES), should be fully defined.
Response: We changed the title to “Gastrointestinal Tract Stabilized Protein Delivery using Disulfide Thermostable Exoshell System” as advised. Please refer to page 1, lines 2-3.
2- The novelty of the study should be highlighted compared to the author’s previous studies, such as Nature communications 2021, 12 (1), 1-15; Nature communications 2017, 8, 1-8.
Response: We have described the novelty of disulphide tES and compared to tES. To better highlight these, we have included the comparison table in supplementary information. Please refer to page 2, 3, 4, 7; lines 71-76, 125-129, 150-154, 265, 286-288; and Table S1.
3- A comparison with the previously published system should be discussed and summarized in a Table.
Response: We have included the comparison between disulphide-tES and normal tES in the manuscript. Please refer to page 7; line 229; and Table S1.
4- References should be updated.
Response: References are updated. Please refer to page 11-14.

Reviewer 2 Report
This study describes the use of a disulfide linked protein nanoparticle for the enhanced proteolytic stability and oral delivery of enzymes. Over all this is a well constructed study and suitable for the publication in the IJMS. However, I suggest the authors to address the following concerns.
1) For the oral delivery of luciferase, I suggest the authors to include the results of tES-nanoparticle along with free Luciferase and DS-tES nanoparticle in Fig 4, as the comparison should be carried out between Disulfide linked and non-linked nanoparticles under in vivo conditions.
Author Response
We thank the reviewer #2 for constructive comment to make the manuscript more readable and appealing to a larger scientific population. The response to the specific comment is given below:
1-For the oral delivery of luciferase, I suggest the authors to include the results of tES-nanoparticle along with free Luciferase and DS-tES nanoparticle in Fig 4, as the comparison should be carried out between Disulfide linked and non-linked nanoparticles under in vivo conditions.
Response: Previously, we have evaluated the effect of Renilla luciferase (rluc) encapsulated within the non-disulfide linked tES (tES-rluc) in oral delivery studies and the luciferase activity was found to be equivalent to the free enzyme (data shown in supplementary information). Since ‘tES’ was inefficient in protecting the enzyme activity from the adverse GI environment, we then proceeded with the design and engineering of DS-tES as an oral delivery vehicle for enzyme/protein delivery. Please refer to page 5, 9; lines 191-192, 339; and Fig. S5.

Reviewer 3 Report
The authors investigated a nanoparticle, i.e. thermostable exoshells (tES), for oral drug delivery applications. They found stability of the disulfide-linked tES (DS-tES) under acidic conditions and against digestive enzymes. A permeability assay in human colon cancer cell line Caco2 and in vivo experiments of oral administration were also carried out. They successfully proved the stability of DS-tES through gastrointestinal tract. The study provides an important contribution to oral drug delivery. However, I have a little concern that need to be addressed.
1. Although most of the presented experiments are well done and appropriate, the description of cell permeability assay was unclear. The authors showed only relative horseradish peroxidase (HRP) activity from the apical side of each filter. The previous report they referred (Hubatsch I. et al. Nature protocols 2007, 2, 2111-2119) described the calculation of permeability from the concentration of apical and basolateral chambers. They should make interpretation of the data appropriate and clear.
2. In the second paragraph of discussion section, the authors mentioned about nanoparticle stability and cellular uptake. Although they revealed the stability of DS-tES successfully both in vitro and in vivo, cellular uptake of ds-tES remains to be established. The paragraph was a little confusing. They should organize the paragraph with description for limitations.

Author Response
We thank the reviewer #3 for constructive comments to make the manuscript more readable and appealing to a larger scientific population. The responses to the specific comments are given below:
1-Although most of the presented experiments are well done and appropriate, the description of cell permeability assay was unclear. The authors showed only relative horseradish peroxidase (HRP) activity from the apical side of each filter. The previous report they referred (Hubatsch I. et al. Nature protocols 2007, 2, 2111-2119) described the calculation of permeability from the concentration of apical and basolateral chambers. They should make interpretation of the data appropriate and clear.
Response: As suggested, we have determined the HRP activity using the formula:
Papp = (dQ/dt) x (1/(AC0))
Further, to validate observations, we have also included the permeability assay results with free and encapsulated Renilla luciferase (rluc). Please refer to page 5, 9; lines 171-175, 333-336; and Fig. S4.
2- In the second paragraph of discussion section, the authors mentioned about nanoparticle stability and cellular uptake. Although they revealed the stability of DS-tES successfully both in vitro and in vivo, cellular uptake of ds-tES remains to be established. The paragraph was a little confusing. They should organize the paragraph with description for limitations.
Response: We agree to R#3 for his constructive comment on cell uptake studies. Currently, we are assessing DS-tES as an oral therapeutic vehicle in mice gastric cancer models. The ongoing study will involve the evaluation of DS-tES in receptor-mediated cellular uptake mechanisms. The findings from this study will be published as a separate manuscript. Additionally, we have also refined the ‘discussion’ with possible limitations as suggested by the reviewer. Please refer to page 7, line 252, 261-265.

Reviewer 4 Report
Sadeghi et al. demonstrated an interesting study by using in vitro, cell and in vivo mouse models showing that after mutating the certain sites identified by the authors, the disulfide-linked tES (DS-tES) exhibited significant intestinal permeability. As the authors stated, DS-tES may potentially be applied for intraluminal and systemic oral drug delivery. Most of the data in the manuscript supported the authors’ conclusions. However, some small issues should be concerned.
1.Figure4 B, the IVIS 3 mice images are spliced in the 3 hours time point. Since these are IVIS images, the authors can not use one color scale bar (which is spliced in figure4B as well) for three different spliced mice images. Same issue for the 24 hours time point IVIS images.
2.In IVIS images, the PBS and renilla luciferase( rluc) groups showed relatively high bioluminescence signals, mainly in the bladder area. In contrast, the DS-tES-rluc group did not show the same bioluminescence pattern. Why does the PBS group show bioluminescence positive signal? Why do the rluc and DS-tES-rluc show different bioluminescence distribution patterns in the bladder area?
Author Response
We thank the reviewer #4 for constructive comments to make the manuscript more readable and appealing to a larger scientific population. The responses to the specific comments are given below:
1- Figure4 B, the IVIS 3 mice images are spliced in the 3 hours’ time point. Since these are IVIS images, the authors cannot use one color scale bar (which is spliced in figure4B as well) for three different spliced mice images. Same issue for the 24 hours’ time point IVIS images.
Response: We have corrected this. Please refer to Figure 4B.
2- In IVIS images, the PBS and renilla luciferase (rluc) groups showed relatively high bioluminescence signals, mainly in the bladder area. In contrast, the DS-tES-rluc group did not show the same bioluminescence pattern. Why does the PBS group show bioluminescence positive signal? Why do the rluc and DS-tES-rluc show different bioluminescence distribution patterns in the bladder area?
Response: In the present study, we used the highly sensitive luciferase substrate- ViviRen. ViviRen contain small protecting groups that when cleaved by intracellular esterases and lipases provide strong luminescence signal upon activation by Renilla luciferase. Since ViviRen was i.p. administered, we observed signal at the site of administration in the abdominal area, presumably due to substrate accumulation and autoluminescence [1]. We have now corrected the background noise to better represent the images. Please refer to Fig. 4. Also, the difference in the bioluminescence pattern can be attributed to the fact that the area under luminescence is wider in DS-tES when compare to the free enzyme.
Reference:
- Otto-Duessel M, Khankaldyyan V, Gonzalez-Gomez I, Jensen MC, Laug WE, Rosol M. In vivo testing of Renilla luciferase substrate analogs in an orthotopic murine model of human glioblastoma. Mol Imaging. 2006;5(2):57-64

Round 2
Reviewer 1 Report
The authors addressed most of the comments and the revised version can be accepted.
Author Response
Thanks for reviewing